# Pre-CRRT furosemide and mortality in sepsis-associated AKI: A retrospective cohort study

**Si-Ye Shen**◉, **Lu Fu**◉, **Dai-Yun Liang, Shao-Kang Wang, Hu Chen**◉, **Li-Jun Cao, Wei-Li Yu, Yun Sun, Zhong-Hua Lu**◉*

The First Department of Critical Care Medicine, The Second Affiliated Hospital of Anhui Medical University, Hefei, Anhui, China

◉ These authors contributed equally to this work.
* luzhonghua077@126.com

## Abstract

### Objective

We aimed to assess whether pre-CRRT furosemide use in SAKI patients requiring CRRT is associated with 28-day mortality, and to evaluate its relationship with secondary outcomes including short-term mortality, in-hospital death, and length of stay. We also examined potential effect modification by CKD status and furosemide dosage.

### Design

Retrospective cohort study.

### Setting

Data were extracted from the MIMIC-IV database, a large publicly available critical care database.

### Participants

A total of 969 adult patients with SAKI requiring CRRT were included.

### Intervention

Patients were stratified based on pre-CRRT furosemide use (defined as administration within 72 hours prior to CRRT initiation). Propensity score matching (1:1) was applied to generate balanced cohorts (n = 560).

**Data availability statement:** The data used in this study were sourced from the MIMIC-IV database (version 2.2), a publicly available resource hosted on the PhysioNet platform (https://physionet.org/content/mimiciv/2.2/). Interested researchers may directly apply for access through the official PhysioNet website (https://physionet.org/content/mimiciv/).

**Funding:** Anhui Provincial Health and Health Research Project (Grant No. 2024Aa20292), the Anhui Province Traditional Chinese Medicine Inheritance and Innovation Research Project (Grant No. 2024CCCX128), the Special Research Fund for Analgesia and Sedation in Critical Care (Grant No. AHEBM20250707M1), and the Natural Science Research Project of Anhui Higher Education Institutions (Grant No. 2023AH053168). The funders had no role in study design, data collection and analysis, decision to publish, or preparation of the manuscript.

**Competing interests:** The authors have declared that no competing interests exist.

## Measurements

Primary outcome: 28-day all-cause mortality. Secondary outcomes: 7-day mortality, 90-day mortality, in-hospital mortality, and length of ICU and hospital stay. Multivariable Cox regression was used to adjust for potential confounders.

## Results

In the matched analysis, furosemide use was associated with significantly lower 28-day mortality (44.3% vs 58.6%; HR 0.58, 95% CI 0.46–0.73, $p < 0.001$). Consistent reductions were observed in 7-day mortality (HR 0.45), 90-day mortality (HR 0.59), and in-hospital mortality (HR 0.50; all $p < 0.01$). Subgroup analyses showed greater benefit in non-chronic kidney disease patients (interaction p = 0.021) and with high-dose furosemide (>90mg/72h; HR 0.62, 95% CI 0.45–0.85). Sensitivity analyses in the full cohort confirmed robustness (HR 0.59, 95% CI 0.49–0.72). The use of furosemide is related to the prolongation of CRRT startup time (1.52 vs 1.46 days). However, furosemide use was associated with prolonged median ICU stay (9.3 vs 6.1 days) and hospital stay (20.8 vs 12.5 days).

## Conclusions

These findings suggest that, in this selected cohort, pre-CRRT furosemide use was associated with lower mortality and does not appear to be associated with harm from delayed CRRT initiation. However, due to the observational design, these results should be warrant confirmation in prospective, randomized controlled trials.

## Introduction

Acute kidney injury (AKI) is a complex syndrome marked by rapid deterioration in renal function, evidenced by elevated serum creatinine levels, decreased urine output, or both [1–4]. Sepsis, characterized by a dysregulated host response to infection that results in life-threatening organ dysfunction [5,6], is a leading cause of AKI in critically ill patients, a condition referred to as sepsis-associated AKI (SAKI) [7]. SAKI is a significant contributor to extended hospital stays, increased mortality rates, and diminished long-term quality of life [7].

In cases of severe SAKI, continuous renal replacement therapy (CRRT) is frequently utilized to manage fluid overload, eliminate inflammatory mediators, and maintain renal and systemic homeostasis [8–10]. Before commencing CRRT, clinicians often administer furosemide to induce diuresis, with the goal of reversing AKI or circumventing the need for invasive therapy [11,12]. The rationale for this practice is supported by several potential mechanisms. By reducing fluid overload, furosemide may alleviate pulmonary edema, improve oxygenation, and facilitate liberation from mechanical ventilation [13]. Conversely, furosemide use carries inherent risks, including electrolyte disturbances and ototoxicity [14]. Although furosemide stress tests have demonstrated prognostic value in early AKI [15], the evidence regarding

its effectiveness in improving clinical outcomes, particularly in patients with SAKI who eventually require CRRT, remains limited and inconclusive [13,16]. Furthermore, it is unclear whether the use of furosemide prior to CRRT initiation might delay necessary intervention and inadvertently lead to worsened patient prognosis.

Given these uncertainties, whether pre-CRRT furosemide use affects prognosis in SAKI patients remains an unresolved clinical question. This study therefore aimed to evaluate this association using real-world data from the MIMIC-IV database.

## Methods

### Study design, participants, and setting

This was a retrospective cohort study using data from the Medical Information Mart for Intensive Care (MIMIC-IV) database, version 2.2. MIMIC-IV contains de-identified electronic health records from 534,912 critical care admissions at Beth Israel Deaconess Medical Center (Boston, MA, USA) between 2008 and 2019.

The demand for CRRT primarily arises in critically ill patients suffering from severe AKI. In light of this clinical context and the inherent limitations in patient distribution within the database, We included critically ill adult patients who (1) met the diagnostic criteria for sepsis and stage 3 AKI; (2) received CRRT during their ICU stay; and (3) were admitted to the ICU for the first time during the study period (for patients with multiple ICU admissions, only the first hospitalization was analyzed). Sepsis was defined according to the Third International Consensus Definitions (Sepsis-3), requiring suspected or confirmed infection accompanied by an acute increase of ≥2 points in the Sequential Organ Failure Assessment (SOFA) score. AKI was staged based on the KDIGO 2012 criteria: an increase in serum creatinine (SCr) of ≥0.3 mg/dL (26.5 µmol/L) within 48 hours, a ≥ 1.5-fold increase from baseline within 7 days, or urine output <0.5 mL/kg/h for ≥6 hours [17].

The exclusion criteria are as follows: (1) age < 18 or >75 years; (2) ICU stay <48 hours; and (3) other diuretics besides furosemide were used.

### Data collection and outcome measures

Data extraction was performed using Structured Query Language (SQL), with scripts sourced from the official MIMIC-IV GitHub repository (https://github.com/MIT-LCP/mimic-iv). Patient characteristics, including age, gender, race, and the Charlson Comorbidity Index, were collected. We extracted instances of furosemide administration within 72 hours prior to CRRT initiation, such as vasopressor use and mechanical ventilation, within 24 hours of ICU admission. Complications were identified using International Classification of Diseases (ICD) coding, which included hypertension, heart failure, diabetes mellitus, chronic kidney disease, chronic obstructive pulmonary disease, and malignancy. Initial ICU admission records included disease severity scores (Acute Physiology and Chronic Health Evaluation [APACHE] II, Sequential Organ Failure Assessment [SOFA] and Charlson Comorbidity Index), vital parameters (heart rate, respiratory rate, mean arterial pressure, SpO2, temperature), and laboratory values (white blood cell count, platelets, albumin, creatinine, lactate, pH, blood urea nitrogen, potassium, sodium, calcium, phosphate).

The exposure variable was pre-CRRT furosemide use, defined as any furosemide administration within 72 hours prior to CRRT initiation, without dosage or route restrictions. The primary outcome was 28-day all-cause mortality. Secondary outcomes included 7-day all-cause mortality, 90-day all-cause mortality, in-hospital mortality, length of ICU stay, and length of hospital stay.

### Statistical analysis

This retrospective analysis lacked a pre-specified statistical analysis plan or sample size calculation, as the cohort was defined by available database entries. Patients were categorized based on pre-CRRT furosemide use. Missing data were handled using multiple imputation [18], and outliers were replaced with median values after IQR-based detection (**S1 Fig and S1 Table**). Multicollinearity was assessed via VIF (all < 5; **S2 and S3 Figs**, **S2 and S3 Tables**).

Continuous variables were reported as mean ± SD or median (IQR) and compared using t-tests or Mann-Whitney U-tests, as appropriate. Categorical variables were presented as counts (%) and analyzed with $\chi^2$ or Fisher's exact tests. Time-to-event outcomes were evaluated with Kaplan-Meier curves and log-rank tests [19,20]. Cox proportional hazards models were used to compute hazard ratios (HR) and 95% confidence intervals (CI). The Hodges-Lehmann method estimated median differences (MD) for continuous outcomes [21]. Key clinical severity scores were categorized via K-means clustering for subgroup analyses (S4 Fig and S4 Table) [22].

Propensity score matching (nearest-neighbor algorithm; 1:1 matching; caliper = 0.05) was performed using variables unrelated to urine output or fluid balance, achieving balance (SMD < 0.10) [23]. Matched cohorts were analyzed for primary and secondary outcomes. Multivariable Cox models adjusted for variables with $p < 0.05$ in univariable analysis.

Subgroup analyses were stratified by age, gender, disease severity scores, and comorbidities. Sensitivity analyses were conducted in the full cohort, adjusting for significant univariate predictors. All analyses used R version 4.2.3, with significance set at two-tailed $p < 0.05$.

### Ethical considerations

The institutional review board at BIDMC granted a waiver of informed consent and approved the data sharing protocols (IRB# 2001-P-001699). Database access authorization (certification ID: 55377862) was obtained through the PhysioNet platform by the corresponding investigator. This study adheres to the Strengthening the Reporting of Observational Studies in Epidemiology (STROBE) guidelines for methodological reporting.

## Results

### Patient selection

The patient selection procedure is depicted in Fig 1. Initially, 73,181 records were found. Following the removal of ineligible entries, the ultimate cohort consisted of 969 patients, including 615 (63.5%) with pre-CRRT furosemide use. The matched cohort contained 560 patients, evenly divided into two groups of 280 each.

### Baseline characteristics

Baseline characteristics of the study population before and after propensity score matching are presented in S5 Table and Table 1. Before matching, significant between-group differences existed in multiple variables, including demographics, vital signs, laboratory values, severity scores, and comorbidities, reflecting confounding by indication (S5 Table). After 1:1 propensity score matching, 280 patients with pre-CRRT furosemide use were successfully matched to 280 patients without furosemide use. The matching procedure achieved excellent covariate balance, with absolute standardized mean differences (SMD) below 0.10 for all variables except serum creatinine, which showed a minimal residual imbalance (SMD = 0.10) (Table 1 and S5 Fig). In the matched cohort, the two groups were well-balanced across all measured baseline characteristics, including age, gender, severity scores (SOFA, APACHE II), comorbidities, and key laboratory parameters. The distribution of propensity scores before and after matching is illustrated in S6 Fig.

### Primary outcome

The 28-day all-cause mortality rate in the furosemide group was 44.29% (124/280), compared to 58.57% (164/280) in the non-furosemide group. Fig 2 presents the Kaplan-Meier curves for 28-day all-cause mortality based on pre-CRRT furosemide use in the matched cohort. Both univariate and multivariable Cox regression confirmed a significant association between pre-CRRT furosemide use and reduced 28-day mortality (univariate HR 0.60, 95% CI 0.48–0.76; multivariable HR 0.58, 95% CI 0.46–0.73; both $p < 0.01$) (Table 2).

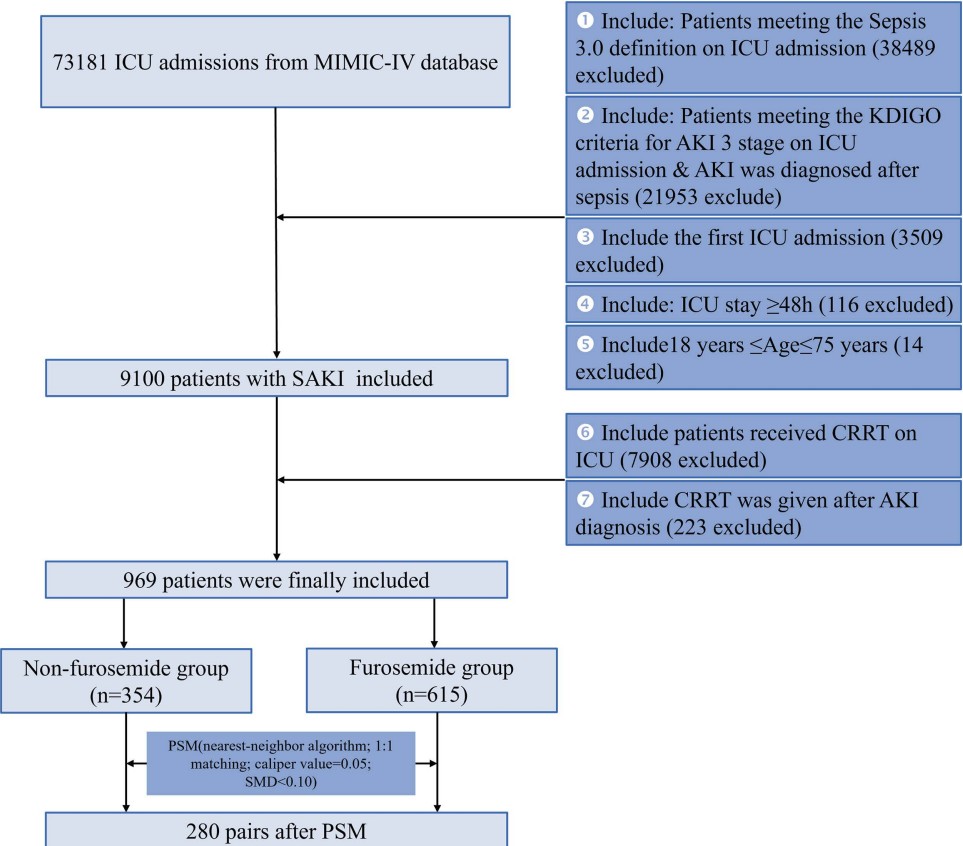

**Fig 1. Flow chart of patient selection.** AKI: Acute kidney injury; CRRT: continuous renal replacement therapy; ICU: intensive care unit; MIMIC-IV: Medical Information Mart for Intensive Care IV; SAKI: Sepsis-associated Acute kidney injury; PSM: propensity score matching.

To assess the robustness of this finding, we performed sensitivity analyses in the full unmatched cohort (N = 969). These analyses yielded consistent results: furosemide use was associated with lower 28-day mortality in both univariate (HR, 0.66; 95% CI, 0.55–0.79; $p < 0.001$) and multivariable models (HR, 0.59; 95% CI, 0.49–0.72; $p < 0.001$) (**S6 and S7 Tables**). The corresponding Kaplan-Meier curve is shown in **S7 Fig**.

### Secondary outcomes

**7-day all-cause mortality, 90-day all-cause mortality and in-hospital mortality.** The 7-day all-cause mortality rate was 18.93% (53/280) in the furosemide group, compared to 39.29% (110/280) in the non-furosemide group. This finding was consistent in the multivariate analysis (HR, 0.45; 95% CI, 0.32–0.62; $p < 0.01$). The 90-day all-cause mortality rate was 48.21% (135/280) in the furosemide group versus 61.43% (172/280) in the non-furosemide group. Both univariate analysis (HR, 0.63; 95% CI, 0.50–0.79; $p < 0.01$) and multivariate analysis (HR, 0.59; 95% CI, 0.47–0.74; $p < 0.01$) indicated that the use of furosemide was associated with a reduced 90-day all-cause mortality rate. The in-hospital mortality rate was 40.36% (113/280) in the furosemide group compared to 56.07% (157/280) in the non-furosemide group. In both univariate analysis (HR, 0.53; 95% CI, 0.41–0.67; $p < 0.01$) and multivariate analysis (HR, 0.50; 95% CI, 0.39–0.64; $p < 0.01$), furosemide use was associated with lower in-hospital mortality (**Table 2**).b69d-4fca-b96a-823a7cda2

**Table 1. Baseline characteristics after propensity score matching.**

| Variable | Group | | | _p-value_ | SMD |
|---|---|---|---|---|---|
| | Overall | Non-Furosemide | Furosemide | | |
| N | 560 | 280 | 280 | | |
| Age (years), median (IQR) | 59 (49.75-67) | 59 (49-67) | 60 (50-68) | 0.44 | 0.06 |
| Male (%), median (IQR) | 338 (60.36) | 165 (58.93) | 173 (61.79) | 0.55 | 0.06 |
| Ethnicity (%), median (IQR) | | | | 0.75 | 0.06 |
| White | 299 (53.39) | 145 (51.79) | 154 (55.00) | | |
| Black | 79 (14.11) | 41 (14.64) | 38 (13.57) | | |
| Other | 182 (32.50) | 94 (33.57) | 88 (31.43) | | |
| Height (cm), median (IQR) | 173 (163-178) | 173 (165-178) | 173 (163-178) | 0.95 | 0.01 |
| Weight (kg), median (IQR) | 88 (72.78-101.82) | 88 (73.82-100.85) | 87.3 (71.01-103.96) | 0.65 | 0.04 |
| Heart rate (/bpm), Mean±SD | 95.79±20.89 | 96.25±22.34 | 95.34±19.37 | 0.61 | 0.04 |
| Respiratory rate (/bpm), median (IQR) | 20 (17-24) | 20 (17-24.25) | 20 (17-24) | 0.99 | <0.01 |
| MAP (mmHg), median (IQR) | 74 (65-83) | 74 (64.75-85.25) | 74 (66-82.25) | 0.43 | 0.07 |
| Body temperature (°C), median (IQR) | 36.78 (36.5-37) | 36.78 (36.5-37) | 36.78 (36.5-37.06) | 0.64 | 0.04 |
| SpO2 (%), median (IQR) | 97 (94-100) | 97 (94-100) | 97 (94-100) | 0.68 | 0.03 |
| WBC (×10⁹/L), median (IQR) | 12.5 (7.6-17.3) | 12.5 (6.88-17.92) | 12.5 (8.2-16.83) | 0.97 | <0.01 |
| Albumin (g/dL), median (IQR) | 2.7 (2.3-3.2) | 2.7 (2.3-3.2) | 2.8 (2.3-3.1) | 0.99 | <0.01 |
| Platelet (×10⁹/L), median (IQR) | 136 (82-206.5) | 130 (80-210.5) | 138 (88-201.75) | 0.65 | 0.04 |
| PT (seconds), median (IQR) | 16.2 (14.2-21.1) | 16.2 (14.1-20.62) | 16.2 (14.3-21.42) | 0.82 | 0.02 |
| Lactate (mmol/L), median (IQR) | 2.3 (1.6-3.9) | 2.3 (1.6-3.7) | 2.3 (1.5-4) | 0.90 | 0.01 |
| PH, median (IQR) | 7.31 (7.21-7.39) | 7.31 (7.2-7.39) | 7.31 (7.22-7.38) | 0.61 | 0.04 |
| Creatinine (mg/dL), median (IQR) | 2.9 (1.67-4.5) | 2.95 (1.9-4.6) | 2.7 (1.4-4.32) | 0.23 | 0.10 |
| Urea nitrogen (mg/dL), median (IQR) | 37 (24-59) | 37 (24-58) | 37 (22.75-59.25) | 0.56 | 0.05 |
| Calcium (mg/dL), median (IQR) | 8.1 (7.3-8.7) | 8.1 (7.3-8.83) | 8.1 (7.3-8.6) | 0.38 | 0.07 |
| Potassium (mEq/L), median (IQR) | 4.4 (3.9-5) | 4.4 (3.9-5.03) | 4.4 (3.8-5) | 0.58 | 0.05 |
| Sodium(mEq/L), median (IQR) | 137 (133-140.25) | 137 (133-140) | 137 (132-141) | 0.78 | 0.02 |
| Phosphate (mg/dL), median (IQR) | 5 (3.8-6.8) | 5 (3.9-6.8) | 5.1 (3.7-6.8) | 0.50 | 0.06 |
| Total urine output[a] (ml), median (IQR) | 420 (100.75-905.75) | 250 (66.5-656.75) | 447 (180-1110.75) | <0.01 | 0.37 |
| Liquid input[b] (ml), median (IQR) | 4746.42 (2781.66-8588.81) | 4979.09 (3025.6-8938.27) | 4650.92 (2636.14-8249.38) | 0.10 | 0.14 |
| Liquid output[b] (ml), median (IQR) | 855 (309.25-1545.25) | 650 (245-1200.5) | 913 (407.5-1910) | <0.01 | 0.24 |
| Fluid balance[b] (ml), median (IQR) | 3553.14 (1528.56-6890.76) | 4199.3 (2030.81-7482.69) | 3318.01 (1049.21-6332.05) | <0.01 | 0.24 |
| Hypertension, n (%) | 129 (23.04) | 63 (22.50) | 66 (23.57) | 0.84 | 0.03 |
| CKD, n (%) | 139 (24.82) | 66 (23.57) | 73 (26.07) | 0.56 | 0.06 |
| Cancer, n (%) | 44 (7.86) | 20 (7.14) | 24 (8.57) | 0.64 | 0.05 |
| Heart failure, n (%) | 155 (27.68) | 80 (28.57) | 75 (26.79) | 0.71 | 0.04 |
| COPD, n (%) | 77 (13.75) | 34 (12.14) | 43 (15.36) | 0.33 | 0.09 |
| Diabetes, n (%) | 196 (35.00) | 99 (35.36) | 97 (34.64) | 0.93 | 0.01 |
| SOFA, median (IQR) | 11 (9-14) | 12 (9-15) | 11 (9-14) | 0.65 | 0.04 |
| APACHEII, median (IQR) | 28 (23-33) | 29 (23-33) | 27 (23-33) | 0.58 | 0.05 |
| CCI, median (IQR) | 5 (4-8) | 5.5 (4-7) | 5 (4-8) | 0.33 | 0.08 |
| Ventilation[c], n (%) | 445 (79.46) | 221 (78.93) | 224 (80.00) | 0.83 | 0.03 |
| Vasopressors[c], n (%) | 222 (39.64) | 107 (38.21) | 115 (41.07) | 0.55 | 0.06 |

APACHEII: Acute Physiology and Chronic Health Evaluation II score; CKD: Chronic kidney disease; COPD: Chronic Obstructive Pulmonary Disease; CCI: Charlson Comorbidity Index; MAP: Mean arterial pressure; PT: Prothrombin Time; SOFA: Sequential Organ Failure Assessment score; WBC: White blood cell.

[a]Total urine output in the 72 hours prior to CRRT.

[b]The most recent fluid intake and output record and fluid balance record before CRRT.

[c]Requirement of vasoactive drugs or mechanical ventilation on the first day of ICU admission.

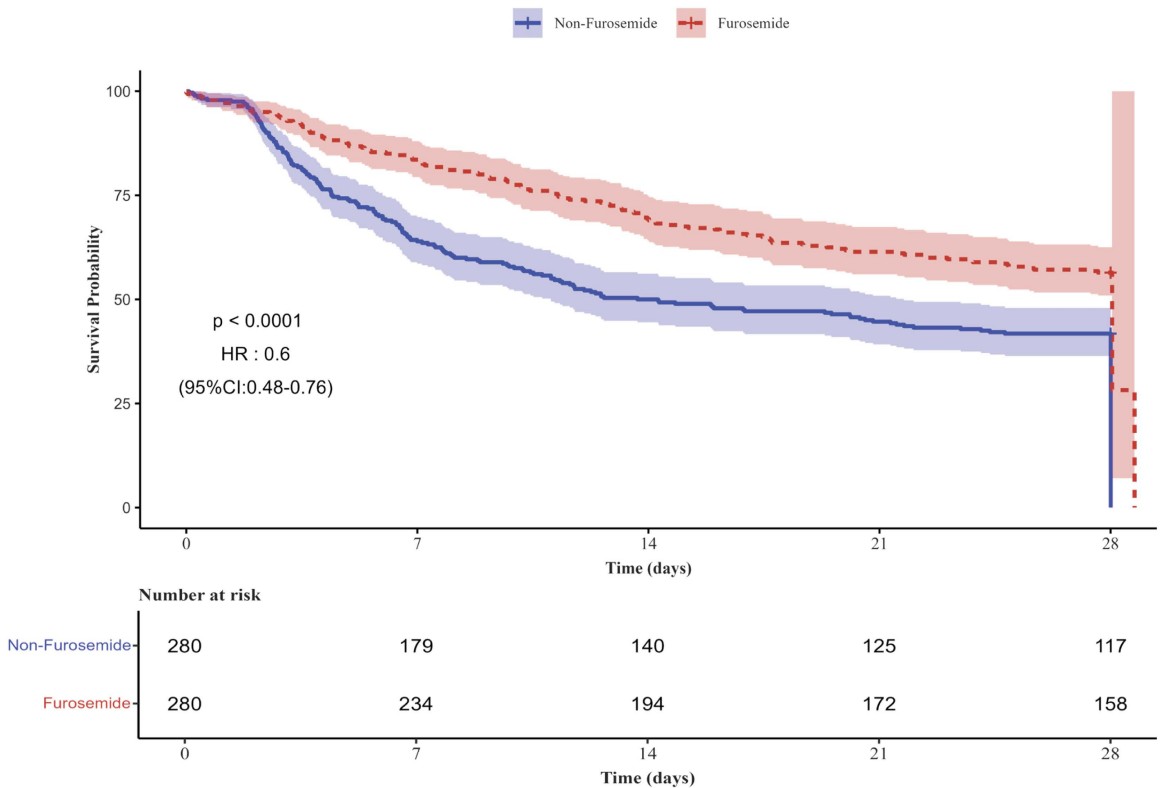

**Fig 2. Kaplan-Meier curve for 28-day all-cause mortality according to furosemide use before 72 hours of CRRT in the matched cohort.** CI: confidence interval; HR: hazard ratio.

**Length of ICU stay and length of hospital stay.** The median length of ICU stay was 9.3 days (IQR 4.10–16.65) for the furosemide use group, compared to 6.11 days (IQR 3.08–11.81) for the no use group. The median length of hospital stay was 20.83 days (IQR 13.09–34.58) in the furosemide use group, whereas it was 12.54 days (IQR 6.07–25.54) in the no use group. Furosemide use was associated with a prolonged length of ICU stay (MD, 2 days; 95% CI, 0.92–3.24; $p < 0.01$) and an extended length of hospital stay (MD, 6.89 days; 95% CI, 4.63–9.26; $p < 0.01$) (**Table 2**).

## Subgroup analyses

**Fig 3** displays the results of subgroup analysis for 28-day all-cause mortality in the matched cohort. In patients without pre-existing CKD, pre-CRRT furosemide use was associated with a lower mortality (HR, 0.53; 95% CI, 0.43–0.66; $p < 0.001$). The $p$-value for interaction between furosemide use and CKD status was 0.021. For all other subgroups examined—including age, sex, SOFA score, APACHE II score, and other comorbidities—the HR point estimates for furosemide use were also below 1.00, with the $p$-values for interaction exceeding 0.05.

## Dosage and prognosis

**S8 Fig** demonstrates a significantly improved 28-day survival rate in patients receiving high-dose furosemide (>90 mg/72h) prior to the initiation of CRRT, as evidenced by Kaplan-Meier analysis (log-rank $p = 0.0023$). Furthermore, multivariable Cox regression analysis confirmed a 38% reduction in mortality associated with high-dose therapy (HR 0.62, 95% CI 0.45–0.85) when compared to low-dose regimens (≤90 mg/72h).

 

**Table 2. The association of Furosemide use before 72 hours of CRRT with outcomes in the matched cohort.**

| Outcomes | Non-Furosemide (n = 280) | Furosemide (n = 280) | Univariable analysis | | Multivariable analysis | |
|---|---|---|---|---|---|---|
| | | | HR/MD (95% CI) | *p*-value | HR/MD (95% CI) | *p*-value |
| Primary outcome | | | | | | |
| 28-day all-cause mortality, n (%) | 164 (58.57) | 124 (44.29) | 0.60 (0.48-0.76) | <0.01 | 0.58 (0.46-0.73) | <0.01 |
| Secondary outcomes | | | | | | |
| 7-day all-cause mortality, n (%) | 110 (39.29) | 53 (18.93) | 0.46 (0.33-0.64) | <0.01 | 0.45 (0.32-0.62) | <0.01 |
| 90-day all-cause mortality, n (%) | 172 (61.43) | 135 (48.21) | 0.63 (0.50-0.79) | <0.01 | 0.59 (0.47-0.74) | <0.01 |
| Death in hospital, n (%) | 157 (56.07) | 113 (40.36) | 0.53 (0.41-0.67) | <0.01 | 0.50 (0.39-0.64) | <0.01 |
| Length of hospital stay (days), median (IQR) | 12.54 (6.07-25.54) | 20.83 (13.09-34.58) | 6.89 (4.63-9.26) | <0.01 | / | / |
| Length of ICU stay (days), median (IQR) | 6.11 (3.08-11.81) | 9.3 (4.10-16.65) | 2 (0.92-3.24) | <0.01 | / | / |

HR: Hazard ratio; MD: Median Difference.

| Subgroups | Count (%) | Non-Furosemide | Furosemide | HR (95% CI) | | P | P for interaction |
|---|---|---|---|---|---|---|---|
| | | No. of events/ No. of total | | | | | |
| All patients | 969 (100.00) | 201/354 | 283/615 | 0.59 (0.49 ~ 0.71) | | <.001 | |
| Age group | | | | | | | 0.506 |
| < 65 yr | 630 (65.02) | 139/244 | 169/386 | 0.56 (0.44 ~ 0.71) | | <.001 | |
| ≥ 65 yr | 339 (34.98) | 62/110 | 114/229 | 0.62 (0.45 ~ 0.85) | | 0.004 | |
| SOFA score | | | | | | | 0.936 |
| Low | 438 (45.20) | 68/134 | 133/304 | 0.52 (0.38 ~ 0.72) | | <.001 | |
| High | 531 (54.80) | 133/220 | 150/311 | 0.62 (0.49 ~ 0.79) | | <.001 | |
| APACHEII score | | | | | | | 0.199 |
| Low | 491 (50.67) | 87/157 | 144/334 | 0.50 (0.38 ~ 0.67) | | <.001 | |
| High | 478 (49.33) | 114/197 | 139/281 | 0.68 (0.53 ~ 0.87) | | 0.003 | |
| Hypertension | | | | | | | 0.064 |
| No | 717 (73.99) | 155/287 | 201/430 | 0.66 (0.53 ~ 0.83) | | <.001 | |
| Yes | 252 (26.01) | 46/67 | 82/185 | 0.40 (0.28 ~ 0.58) | | <.001 | |
| CKD | | | | | | | 0.021 |
| No | 740 (76.37) | 164/273 | 212/467 | 0.53 (0.43 ~ 0.66) | | <.001 | |
| Yes | 229 (23.63) | 37/81 | 71/148 | 0.87 (0.58 ~ 1.32) | | 0.513 | |
| Cancer | | | | | | | 0.609 |
| No | 895 (92.36) | 182/324 | 260/571 | 0.61 (0.50 ~ 0.74) | | <.001 | |
| Yes | 74 (7.64) | 19/30 | 23/44 | 0.50 (0.26 ~ 0.95) | | 0.035 | |
| Heart faliure | | | | | | | 0.432 |
| No | 668 (68.94) | 152/261 | 185/407 | 0.55 (0.44 ~ 0.69) | | <.001 | |
| Yes | 301 (31.06) | 49/93 | 98/208 | 0.65 (0.45 ~ 0.95) | | 0.026 | |
| COPD | | | | | | | 0.193 |
| No | 835 (86.17) | 169/307 | 243/528 | 0.62 (0.51 ~ 0.76) | | <.001 | |
| Yes | 134 (13.83) | 32/47 | 40/87 | 0.39 (0.23 ~ 0.64) | | <.001 | |
| Diabetes | | | | | | | 0.958 |
| No | 622 (64.19) | 140/236 | 187/386 | 0.61 (0.48 ~ 0.76) | | <.001 | |
| Yes | 347 (35.81) | 61/118 | 96/229 | 0.58 (0.41 ~ 0.81) | | 0.002 | |
| Gender | | | | | | | 0.289 |
| Female | 372 (38.39) | 85/148 | 98/224 | 0.50 (0.37 ~ 0.69) | | <.001 | |
| Male | 597 (61.61) | 116/206 | 185/391 | 0.64 (0.50 ~ 0.81) | | <.001 | |

0      1      1.5

**Fig 3. Subgroup analyses for 28-day all-cause mortality in the matched cohort.** APACHEII: Acute Physiology and Chronic Health Evaluation II score; CKD: Chronic kidney disease; COPD: Chronic Obstructive Pulmonary Disease.

## Timing of CRRT initiation

Concerning the timing of CRRT initiation, the median duration from the diagnosis of SAKI to the commencement of CRRT was significantly longer in the furosemide group compared to the non-furosemide group, with values of 1.52 days (IQR 0.88–3.15) versus 1.46 days (IQR 0.58–1.89); $p < 0.01$. This observed difference equates to an approximate delay of 1.44 hours in CRRT initiation associated with the use of furosemide prior to CRRT (Table 3).

## Discussion

In this cohort of SAKI patients requiring CRRT, pre-CRRT furosemide use was associated with significantly lower 28-day mortality—a finding that remained consistent across subgroup and sensitivity analyses. Despite the observed delay in CRRT initiation, this delay was not associated with increased mortality. This finding is reassuring, as it suggests that the practice of delaying CRRT to attempt diuresis does not appear to translate into net harm—at least within the 72-hour window defined in this study. This observation may be explained by the possibility that patients selected for furosemide represent a subgroup with greater renal reserve, and the diuretic trial itself may help identify those with a better underlying prognosis.

These results align with some prior reports but not others. Zhao et al. similarly found furosemide associated with reduced mortality in AKI patients, while Krzych et al. reported no such benefit [12,13]. More recently, Li et al. observed lower in-hospital mortality with furosemide in SAKI patients receiving RRT, consistent with our findings [24]. The hetero-geneity across studies likely reflects differences in patient selection, timing of furosemide administration, and severity of illness. However, due to the severity of most patients undergoing CRRT and the limitations of the database, the study found that stage 3 SAKI patients accounted for an extremely high proportion among those receiving CRRT. To avoid bias, this study only included stage 3 SAKI patients and explored whether furosemide treatment failure within the first 72 hours prior to CRRT could lead to adverse outcomes due to potential delays in CRRT initiation, which differs from the research conducted by Caifeng Li and colleagues. Furthermore, the use of furosemide was associated with lower rates of 7-day all-cause mortality, 90-day all-cause mortality, ICU mortality, and in-hospital mortality. These findings not only corroborate but also expand upon previous observations. Zhao et al. reported in a large cohort of AKI patients from the MIMIC-III database that the use of furosemide was associated with a reduction in both in-hospital and 90-day mortality rates [12]. Similarly, Li et al., in their study of SAKI patients undergoing RRT within the MIMIC-IV database, found that furosemide use correlated with decreased in-hospital mortality [24]. However, data regarding 7-day mortality—outcomes that may more accurately reflect the immediate effects of diuretic therapy and the timing of CRRT—remains limited. Our study provides novel evidence indicating that the use of furosemide prior to CRRT is linked to these early mortality outcomes, thereby reinforcing the potential advantages of this treatment strategy. These findings collectively suggest an association between the use of furosemide and improved outcomes in critically ill patients with sepsis-associated acute kidney injury, even when CRRT is required within 72 hours.

It is noteworthy that this study observed that the furosemide group had longer hospital stays and ICU admission durations. The study by Guang-ju Zhao and colleagues also reported that the use of furosemide was associated with

**Table 3. Differential analysis of time from SAKI diagnosis to initiation of CRRT.**

| Variable | Overall | Non-Furosemide | Group Furosemide | p-value |
|---|---|---|---|---|
| N | 560 | 280 | 280 | |
| Time of AKI to CRRT, (days), median (IQR) | 1.46 (0.75, 2.62) | 1.46 (0.58, 1.89) | 1.52 (0.88, 3.15) | <0.01 |

IQR: Interquartile range.

prolonged ICU and hospital length of stay, which is consistent with our findings [12]. However, the research by Krzych and colleagues found that the use of furosemide did not affect the duration of hospitalization [14]. The prolonged length of stay observed in our study may be partly attributed to the lower mortality rate in the furosemide group, as surviving patients naturally accumulate longer hospital days. Nevertheless, it might also indicate a more complicated clinical course among survivors, warranting further investigation.

Subgroup analysis revealed that patients without concurrent CKD had lower HR values, with interaction P-values less than 0.05, indicating that CKD may modulate the effect of furosemide treatment prior to CRRT. The attenuated benefit in CKD patients may reflect structural renal damage—including tubular atrophy and interstitial fibrosis—which reduces the responsiveness to loop diuretics by impairing sodium reabsorption and NKCC2 cotransporter function [25,26]. Conversely, patients without CKD, who predominantly have acute reversible injury, may derive greater benefit from furosemide-mediated volume optimization prior to CRRT [27–30].

Fluid resuscitation is a critical component of sepsis management, aimed at maintaining hemodynamic stability [31,32]. However, in patients with SAKI, oliguria or anuria may exacerbate fluid accumulation, and aggressive fluid resuscitation frequently results in fluid overload [33,34]. Fluid overload is often significantly associated with adverse outcomes, whereas a lower positive fluid balance, particularly in patients with AKI, is generally linked to reduced ventilator requirements, decreased need for RRT, and improved clinical outcomes, which may help patients through the critical period [35–38]. In medical practice, the diuretic furosemide effectively alleviates fluid overload by altering the patient's urinary status from oliguria to normal urination [16]. Although this medication is not employed as a direct treatment for acute kidney injury (AKI), a study encompassing multiple intensive care units has demonstrated a significant association between the use of such diuretics and improved survival rates in patients with reduced urine output and severe fluid retention [39,40]. Our study indicates that even when furosemide therapy does not achieve expected effects, the furosemide group exhibits lower fluid accumulation and mortality; moreover, higher furosemide dosing correlates with reduced 28-day all-cause mortality in this cohort. This suggests that the use of furosemide reduces fluid accumulation in SAKI patients, and the beneficial effects may outweigh the potential adverse effects of possible CRRT delay. However, existing evidence suggests that the clinical benefits of furosemide are balanced against its potential risks in a dose-dependent manner. High-dose application may significantly increase the risk of adverse events such as electrolyte disturbances, hypovolemia, and worsening renal function [41,42]. Currently, there is still controversy regarding the optimal furosemide dose threshold for SAKI patients and its impact on long-term prognosis. Multicenter randomized controlled trials based on phenotypic stratification are urgently needed to further elucidate the dose-effect relationship. However, existing evidence suggests that the clinical benefits of furosemide are balanced against its potential risks in a dose-dependent manner. High-dose application may significantly increase the risk of adverse events such as electrolyte disturbances, hypovolemia, and worsening renal function [43]. The observed dose-response relationship in our study may therefore reflect a composite of different administration strategies, and we cannot exclude the possibility that the association between dose and outcome differs by infusion modality. Future studies with granular data on administration route and duration are needed to clarify whether the method of furosemide delivery modifies its effect on mortality in SAKI patients.

## Limitations

Several limitations of this study should be acknowledged. First, despite propensity score matching and multivariate adjustment, residual confounding may persist due to the observational nature of this study. Notably, even after matching, clinically meaningful differences in urine output and fluid balance were observed between the furosemide and non-furosemide groups. These variables present a methodological challenge: they serve both as mediators of furosemide's therapeutic effect and as markers of underlying renal reserve and illness severity. Adjusting for them in outcome models could introduce overadjustment bias by 'conditioning on an intermediate,' potentially obscuring the true total effect of furosemide. Conversely, not accounting for them leaves open the possibility that the observed mortality association is partly explained by these residual differences, which may reflect unmeasured factors such as intrinsic renal

responsiveness, adequacy of resuscitation, or subtle differences in tubular injury. Therefore, while our primary analysis employed PSM and multivariable Cox regression on a priori confounders, we cannot exclude the possibility that some residual confounding—particularly related to these physiologically intertwined variables—influences our findings. Second, the generalizability of our findings may be limited to critically ill SAKI patients with advanced AKI (stage 3) requiring CRRT. Finally, our findings demonstrate an association, not causation, and should not be interpreted as evidence of a direct beneficial effect of furosemide. The detailed mechanisms underlying the observed association require further basic and clinical investigation.

## Clinical implication

While causality cannot be inferred from this observational study, the findings offer several provisional insights for clinicians. First, in patients with SAKI being considered for CRRT, a trial of furosemide—when not contraindicated—may be reasonable, as it was not associated with harm even among those who ultimately required RRT. Second, the more pronounced benefit observed in patients without pre-existing CKD suggests that furosemide responsiveness may help identify individuals with greater renal reserve who could potentially avoid or delay CRRT. Third, higher furosemide doses (>90 mg/72h) were associated with greater mortality reduction, although this must be balanced against the known dose-dependent risks of electrolyte disturbances and ototoxicity. Importantly, these findings should not be interpreted as supporting routine high-dose furosemide, but rather as hypothesis-generating observations that require confirmation in randomized trials. Pending such evidence, clinical judgment regarding volume status, hemodynamic stability, and individual patient characteristics should guide pre-CRRT diuretic use.

## Conclusions

In conclusion, this retrospective study suggests that pre-CRRT furosemide use was associated with lower mortality in critically ill patients with SAKI. The findings provide reassurance that furosemide use prior to CRRT, even if it delays initiation, may not be associated with worse outcomes, and could potentially identify a subgroup of patients with a more favorable prognosis. The benefits appear more pronounced in patients without pre-existing CKD and with higher furosemide doses. Given the inherent limitations of the observational design, these findings warrant validation in prospective, randomized controlled trials to establish causality and guide clinical practice.

## Supporting information

**S1 Fig. Percentage of missing data of each variable.** Abbreviations: MAP: Mean arterial pressure; WBC: White blood cell; PT: Prothrombin Time; CKD: Chronic kidney disease; COPD: Chronic Obstructive Pulmonary Disease; SOFA: Sequential Organ Failure Assessment score; APACHEII: Acute Physiology and Chronic Health Evaluation II score; CCI: Charlson Comorbidity Index.
(TIF)

**S2 Fig. Variance inflation factor of each variable in the matched cohort.** Abbreviations: WBC: White blood cell; PT: Prothrombin Time; SOFA: Sequential Organ Failure Assessment score; APACHEII: Acute Physiology and Chronic Health Evaluation II score; A variance inflation factor of <5 for each variable suggested the absence of multicollinearity.
(TIF)

**S3 Fig. Variance inflation factor of each variable in the entire cohort.** Abbreviations: WBC: White blood cell; PT: Prothrombin Time; SOFA: Sequential Organ Failure Assessment score; APACHEII: Acute Physiology and Chronic Health Evaluation II score; A variance inflation factor of <5 for each variable suggested the absence of multicollinearity.
(TIF)

**S4 Fig. K-Means clustering analysis clustering scatter plot.** Abbreviations: SOFA: Sequential Organ Failure Assessment score; APACHEII: Acute Physiology and Chronic Health Evaluation II score; Charlson score: Charlson Comorbidity Index. (TIF)

**S5 Fig. Distributional balance before and after propensity score matching Standardised mean differences in before and after propensity score matching.** Abbreviations: MAP: Mean arterial pressure; WBC: White blood cell; PT: Prothrombin Time; CKD: Chronic kidney disease; COPD: Chronic Obstructive Pulmonary Disease; SOFA: Sequential Organ Failure Assessment score; APACHEII: Acute Physiology and Chronic Health Evaluation II score; Ethnicity_1.0: White; Ethnicity_2.0: Black; Ethnicity_3.0: Other. (TIF)

**S6 Fig. Distributional balance before and after propensity score matching.** (TIF)

**S7 Fig. Kaplan-Meier curve for 28-day all-cause mortality according to furosemide use before 72 hours of CRRT in the unmatched cohort.** (TIF)

**S8 Fig. Kaplan-Meier curve for 28-day all-cause mortality according to dose of Furosemide before 72 hours of CRRT.** (TIF)

**S9 Fig. Pre-CRRT furosemide and mortality in sepsis-associated AKI: A retrospective cohort study.** This retrospective cohort study utilized the MIMIC-IV 2.2 database to enroll 969 patients with SAKI who were first admitted to the ICU, aged between 18 and 75 years, had a hospital stay of 48 hours or more, and required CRRT. Through PSM, 280 patients who received furosemide within 72 hours prior to CRRT were paired with 280 patients who initiated CRRT directly without the use of furosemide. The primary outcome measured was 28-day mortality (HR = 0.58, 95% CI 0.46–0.73), while the secondary outcome was in-hospital mortality (HR = 0.50, 95% CI 0.39–0.64). The Kaplan-Meier curve illustrated a lower 28-day all-cause mortality rate in the furosemide group. In conclusion, within the selected cohort, the use of furosemide prior to CRRT was associated with lower mortality, with no observed harm from a delay in CRRT initiation. Abbreviations: AKI: Acute kidney injury; SAKI: Sepsis-associated acute kidney injury; CRRT: Continuous renal replacement therapy; ICU: Intensive care unit. (TIF)

**S1 Table. Percentage of missing data of each variable.** Abbreviations: MAP: Mean arterial pressure; WBC: White blood cell; PT: Prothrombin Time; CKD: Chronic kidney disease; COPD: Chronic Obstructive Pulmonary Disease; SOFA: Sequential Organ Failure Assessment score; APACHEII: Acute Physiology and Chronic Health Evaluation II score; CCI: Charlson Comorbidity Index. (DOCX)

**S2 Table. Variance inflation factor of each variable in the matched cohort.** Abbreviations: WBC: White blood cell; PT: Prothrombin Time; SOFA: Sequential Organ Failure Assessment score; APACHEII: Acute Physiology and Chronic Health Evaluation II score; A variance inflation factor of <5 for each variable suggested the absence of multicollinearity. (DOCX)

**S3 Table. Variance inflation factor of each variable in the entire cohort.** Abbreviations: WBC: White blood cell; PT: Prothrombin Time; SOFA: Sequential Organ Failure Assessment score; APACHEII: Acute Physiology and Chronic Health Evaluation II score; A variance inflation factor of <5 for each variable suggested the absence of multicollinearity. (DOCX)

**S4 Table. K-Means clustering analysis.** Abbreviations: SOFA: Sequential Organ Failure Assessment score; APACHEII: Acute Physiology and Chronic Health Evaluation II score; Charlson score: Charlson Comorbidity Index.
(DOCX)

**S5 Table. Baseline characteristics before propensity score matching.** Abbreviations: MAP: Mean arterial pressure; WBC: White blood cell; PT: Prothrombin Time; CKD: Chronic kidney disease; COPD: Chronic Obstructive Pulmonary Disease; SOFA: Sequential Organ Failure Assessment score; APACHEII: Acute Physiology and Chronic Health Evaluation II score; CCI: Charlson Comorbidity Index. [a] Total urine output in the 72 hours prior to CRRT. [b] The most recent fluid intake and output record and fluid balance record before CRRT. [c] Requirement of vasoactive drugs or mechanical ventilation on the first day of ICU admission.
(DOCX)

**S6 Table. The Univariable Cox regression results of 28-day all-cause mortality for the use of furosemide within 72 hours prior to CRRT in the unmatched cohort.** Abbreviations: MAP: Mean arterial pressure; WBC: White blood cell; PT: Prothrombin Time; CKD: Chronic kidney disease; COPD: Chronic Obstructive Pulmonary Disease; SOFA: Sequential Organ Failure Assessment score; APACHEII: Acute Physiology and Chronic Health Evaluation II score; Charlson: Charlson Comorbidity Index; HR: Heart rate; RR: Respiratory rate.
(DOCX)

**S7 Table. The multivariate Cox regression results of 28-day all-cause mortality for the use of furosemide within 72 hours prior to CRRT in the unmatched cohort.** Abbreviations: WBC: White blood cell; PT: Prothrombin Time; SOFA: Sequential Organ Failure Assessment score; APACHEII: Acute Physiology and Chronic Health Evaluation II score.
(DOCX)

## Acknowledgments

Thanks to the experimental Center of the Scientific Research Department and the First Department of Critical Care Medicine of the Second Affiliated Hospital of Anhui Medical University for the help of this study. Thank you for the Medical Information Mart for Intensive Care (https://www.home-for-researchers.com).

## Author contributions

**Conceptualization:** Si-Ye Shen, Lu Fu, Li-Jun Cao, Yun Sun, Zhong-hua Lu.

**Data curation:** Si-Ye Shen.

**Funding acquisition:** Zhong-hua Lu.

**Investigation:** Dai-Yun Liang, Shao-Kang Wang, Hu Chen.

**Methodology:** Si-Ye Shen, Lu Fu, Hu Chen.

**Project administration:** Dai-Yun Liang, Shao-Kang Wang, Hu Chen.

**Software:** Si-Ye Shen.

**Supervision:** Dai-Yun Liang, Shao-Kang Wang, Hu Chen, Zhong-hua Lu.

**Writing – original draft:** Si-Ye Shen, Lu Fu, Dai-Yun Liang, Hu Chen, Yun Sun, Zhong-hua Lu.

**Writing – review & editing:** Si-Ye Shen, Wei-Li Yu, Yun Sun, Zhong-hua Lu.

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
