## [Decision Letter · Decision Letter 0]

14 Jan 2026

Dear Dr. Lu,

Thank you for submitting your manuscript to PLOS ONE. After careful consideration, we feel that it has merit but does not fully meet PLOS ONE’s publication criteria as it currently stands. Therefore, we invite you to submit a revised version of the manuscript that addresses the points raised during the review process.

We look forward to receiving your revised manuscript.

Kind regards,

Tarek Samy Abdelaziz, MD,FRCP

Academic Editor

PLOS One

Journal Requirements:

“This work was supported by the Anhui Provincial Health and Health Research Project (2024Aa20292), the Anhui Province Traditional Chinese Medicine Inheritance and Innovation Research Project (2024CCCX128), the Special Research Fund for Analgesia and Sedation in Critical Care (AHEBM20250707M1), and the Natural Science Research Project of Anhui Higher Education Institutions (2023AH053168).”

The author(s) received no specific funding for this work.”

“This work was supported by the Anhui Provincial Health and Health Research Project (2024Aa20292), the Anhui Province Traditional Chinese Medicine Inheritance and Innovation Research Project (2024CCCX128), the Special Research Fund for Analgesia and Sedation in Critical Care (AHEBM20250707M1), and the Natural Science Research Project of Anhui Higher Education Institutions (2023AH053168).”

6. In the online submission form you indicate that your data is not available for proprietary reasons and have provided a contact point for accessing this data. Please note that your current contact point is a co-author on this manuscript. According to our Data Policy, the contact point must not be an author on the manuscript and must be an institutional contact, ideally not an individual. Please revise your data statement to a non-author institutional point of contact, such as a data access or ethics committee, and send this to us via return email. Please also include contact information for the third party organization, and please include the full citation of where the data can be found.

7. Please include a separate caption for each figure in your manuscript.

8. Please include your tables as part of your main manuscript and remove the individual files. Please note that supplementary tables (should remain/ be uploaded) as separate "supporting information" files.

Reviewers' comments:

Reviewer's Responses to Questions

**Comments to the Author**

1. Is the manuscript technically sound, and do the data support the conclusions?

Reviewer #1: Yes

Reviewer #2: Yes

2. Has the statistical analysis been performed appropriately and rigorously?

Reviewer #1: Yes

Reviewer #2: Yes

3. Have the authors made all data underlying the findings in their manuscript fully available?

Reviewer #1: Yes

Reviewer #2: Yes

4. Is the manuscript presented in an intelligible fashion and written in standard English?

Reviewer #1: Yes

Reviewer #2: Yes

Reviewer #1: This is a well-designed retrospective cohort study addressing an important clinical dilemma in the management of sepsis-associated AKI. The large sample size, consistent mortality findings across multiple analyses, and dose-response exploration are notable strengths. The manuscript contributes valuable real-world evidence to an area with limited data.

Major Points:

Causality and Interpretation

The conclusions should be tempered to emphasize association rather than benefit. Statements implying safety or effectiveness should be clearly framed within the limitations of observational data.

Residual Confounding

Although propensity score matching was performed, residual differences in urine output and fluid balance remain clinically meaningful. These variables may reflect both treatment response and baseline severity and should be discussed more explicitly as potential mediators or confounders.

Data Availability Compliance

Please revise the Data Availability Statement to comply with PLOS ONE policy by explicitly stating that the data are derived from the MIMIC-IV database and providing access details.

CRRT Timing

Since one of the main concerns addressed by the study is delayed CRRT initiation, providing more granular data or discussion on time-to-CRRT would further strengthen the manuscript.

Minor Points:

Improve language clarity and reduce repetition, especially in the Discussion.

Ensure consistent terminology (e.g., “pre-CRRT furosemide use” vs “furosemide administration within 72 hours”).

Consider adding a brief clinical implication section to guide readers on how findings may inform practice without overstating conclusions.

Reviewer #2: This is a very good and novel study that addresses an important clinical question. The study is well designed, methodologically sound, and provides valuable insights with potential implications for clinical practice.

Comment 1:

Score Matching

Line 2-3: Very interesting title, but you don’t need to mention the statistical tests used in the study (Propensity Score Matching).

Comment 2:

Line 24-29:

Specify the precise objectives of your study; currently, you only state the aim.

Comment 2:

Line 76:

The introduction is very well written, clear, and effectively aligned with the study's core objectives. I suggest adding a paragraph that discusses the potential mechanisms by which pre-CRRT furosemide administration might influence mortality, to strengthen the clinical rationale and contextualize the study findings.

Line 82:

I recommend adding a dedicated subsection titled *“Study design, participants, and setting.” Additionally, the study design is not clearly described and should be explicitly stated to improve clarity and methodological transparency.

Line 94:

Include this criterion within the inclusion criteria (Line 87).

Line 136:

For Figure 1, the propensity score matching flow diagram is conceptually appropriate and consistent with a 1:1 matching approach. However, the figure lacks information on excluded unmatched patients and should be complemented by a clear description of the matching methodology and balance diagnostics to ensure transparency and reproducibility.

Line 140:

Please revise the manuscript to report baseline characteristics for the propensity score–matched cohort and present them after propensity score matching, as this approach is essential for reducing bias and controlling for confounding. Baseline characteristics before matching may also be reported; however, the primary focus should be on the post-matching results.

Line 176:

3.5. Subgroup analyses:

Please report the results directly with their statistical significance. Avoid interpreting confidence intervals or p-values. The Results section should focus on presenting the findings rather than discussing their statistical implications.

Line 185:

3.6. Sensitivity analyses:

I believe there is redundancy between this section and lines 154–158, which makes the presentation confusing. Please clarify how this section differs from lines 154–158 or consider consolidating the overlapping content to improve clarity, or combine with primary outcome paragraph.

Line 192:

3.7. Dosage and Prognosis:

Please clarify the furosemide dosing strategy, specifically whether it was administered as a continuous infusion over 24 hours or as intermittent dosing. This distinction is important, as it remains an area of debate in the literature and may influence clinical outcomes.

Line 213 – 216:

Please indicate whether there are previous studies worldwide that have assessed these outcomes (7-day, 90-day, ICU, and in-hospital mortality) in relation to furosemide use, and briefly contextualize the findings in relation to your findings.

.

Reviewer #1: **Yes:** Elabbass Ali AbdelmahmuodElabbass Ali AbdelmahmuodElabbass Ali AbdelmahmuodElabbass Ali Abdelmahmuod

Reviewer #2: **Yes:** Mohamed A. AlbekeryMohamed A. AlbekeryMohamed A. AlbekeryMohamed A. Albekery

---

## [Author Response · Author response to Decision Letter 1]

2 Mar 2026

We would like to express our gratitude to all editors and reviewers for their valuable comments and suggestions. In response, we have meticulously revised the original manuscript to enhance clarity and improve readability for our audience. A detailed point-by-point response to the reviewers' comments and concerns can be found in the blue section of the Response to Reviewers document. All page numbers correspond to the document of "Revised Manuscript with Track Changes". We hope that this revision adequately addresses the feedback provided by the editors and reviewers.

---

## [Editor Report · Decision Letter 1]

30 Mar 2026

Pre-CRRT furosemide and mortality in sepsis-associated AKI: A retrospective cohort study

PONE-D-25-46945R1

Dear Dr. Lu,

We’re pleased to inform you that your manuscript has been judged scientifically suitable for publication and will be formally accepted for publication once it meets all outstanding technical requirements.

Kind regards,

Tarek Samy Abdelaziz, MD,FRCP

Academic Editor

PLOS One
---

## [Editor Report · Acceptance letter]

PONE-D-25-46945R1

PLOS One

Dear Dr. Lu,

I'm pleased to inform you that your manuscript has been deemed suitable for publication in PLOS One. Congratulations! Your manuscript is now being handed over to our production team.

Kind regards,

on behalf of

Professor Tarek Samy Abdelaziz

Academic Editor

PLOS One